# Effect of Sample Thickness and Post-Processing on Mechanical Properties of 3D-Printed Titanium Alloy

**DOI:** 10.3390/ma18215008

**Published:** 2025-11-02

**Authors:** Aleš Jíra, Jaroslav Kruis, Zdeněk Tolde, Jan Krčil, Jitřenka Jírů, Jaroslav Fojt

**Affiliations:** 1Department of Mechanics, Faculty of Civil Engineering, Czech Technical University in Prague, 16629 Prague, Czech Republic; jaroslav.kruis@fsv.cvut.cz; 2Department of Physics, Faculty of Mechanical Engineering, Czech Technical University in Prague, 16000 Prague, Czech Republic; zdenek.tolde@fs.cvut.cz (Z.T.); jan.krcil@fs.cvut.cz (J.K.); 3Department of Metals and Corrosion Engineering, Faculty of Chemical Technology, University of Chemistry and Technology, 16628 Prague, Czech Republic; jiruj@vscht.cz (J.J.); fojtj@vscht.cz (J.F.)

**Keywords:** 3D printing, surface etching, annealing, titanium alloy, porosity, mechanical testing

## Abstract

3D printing of beta titanium alloys for biomedical applications is currently in great demand, both for material reasons and for the possibility of producing very complex replacements, often directly tailored to the patient. Gyroidal and similar structures are ideal for biomedical replacements but their manufacturing require specific additive technology and post-processing like annealing or etching. The aim of this work is to determine the mechanical properties of Ti25Nb4Ta8Sn alloy which overcomes Ti6Al4V in biomedical applications. The results showed that Ti6Al4V exhibited a significantly higher ultimate tensile strength (up to 1200 MPa) compared with the beta titanium alloy Ti25Nb4Ta8Sn (up to 360 MPa), while the latter demonstrated a substantially lower elastic modulus (∼40–50 GPa), beneficial for biomedical applications. Annealing improved strength and reduced internal stresses in both alloys, while etching effectively removed residual powder but slightly decreased mechanical integrity. These findings provide a quantitative basis for optimizing printing and post-processing parameters of beta titanium alloys for implant design. The properties will be used for future numerical simulations of implants made from Ti25Nb4Ta8Sn alloy based on discrete particle grid models.

## 1. Introduction

During the last decade, a revolutionary change has taken place in manufacturing driven by 3D printing, which has opened new possibilities for designing and producing high-performance materials. 3D printing enables the creation of intricate and precise structures, making it a critical technology in the biomedical field, offering numerous benefits and innovations. The accuracy of metal 3D printing depends on the manufacturing technology, material, and post-processing, and it can reach levels comparable to conventional machining. Typical tolerances for commonly used methods such as selective laser melting (SLM) are around 0.05 mm. The final dimensional accuracy strongly depends on machine calibration, powder quality, and subsequent machining, which ensures the required precision where necessary. This is particularly crucial in the development of implants and joint replacements, where personalized solutions are essential to accommodate each patient’s unique anatomical characteristics [1,2,3].

Intraosseous implants are most commonly used in the elderly population, in which osteoporosis represents one of the major risk factors affecting implant survival [4] and is also associated with an increased number of revision surgeries [5]. Another considerable factor is the growing prevalence of allergies or hypersensitivity reactions to common metals (particularly CoCrMo, Ti) and bone cements in the population [6,7]. Osteoporosis also alters both the early and late phases of implant fixation as well as bone architecture, leading to fixation loss [8]. In this context, 3D printing introduces new possibilities for crafting complex structures from alloys, offering shape and design options that were previously challenging or impossible with traditional manufacturing methods.

The main efficiency of utilizing 3D printing lies in using complex porous structures with an open pore system, ensuring optimal integration of the implant with surrounding bone tissue. One of the most usable structures is the gyroid structure [9,10]. Inspired by mathematics and geometry, gyroid structures have gained attention in 3D printing due to their distinctive internal architecture. These structures form a network of planar and interconnected supports, creating a three-dimensional grid with high porosity. In biomedical applications, this structure facilitates the easy supply of nutrients and fluids [11], which is vital for tissue regeneration or as a bone replacement.

The gyroid structure can serve as a trabecular or wall-like implant [10,12]. The typical thickness of a wall ranges from 150 to 600 μm, pushing the limits of current printers and the powders used. Numerical simulation of gyroid structures and other wall-like implants is impeded by the lack of data primarily related to material internal porosity and incomplete fusion of powder particles at the edges of structures [13].

Another critical factor for a successful application of a 3D-printed implant is the material used and its biomechanical reliability. Most of the current replacements are made of titanium alloy Ti6Al4V ELI [14], which is validated by medical practice and has been used for a long time. However, it also has disadvantages such as a high Young’s modulus of elasticity or possible toxicity, making alternatives worth considering. A potential solution to the problem of allergic or hypersensitivity reactions to metallic implant materials is the use of advanced Ti-based alloys. Promising compositions include alloys containing Ti, Ta, Zn, Zr, and Nb, which exhibit improved biological properties and a lower elastic modulus compared to the conventional Ti–6Al–4V alloy [15,16]. Among the materials pivotal to this shift are beta titanium alloys, combining titanium’s lightweight nature with excellent durability, strength, thermal stability, biocompatibility, and corrosion resistance [15,16,17,18]. One such alloy is Ti25Nb4Ta8Sn, in which niobium (Nb) and tantalum (Ta) stabilize the titanium in its beta structure [19] (body-centered cubic lattice—BCC/K8). Unfortunately, the high amount of beta titanium stabilizers can lead to increased occurrences of martensitic alpha and omega phases, the latter resulting in reduced fatigue strength or hardness [20]. The addition of tin (Sn) to the alloy prevents these undesirable phenomena. Consequently, such alloys achieve a very low modulus of elasticity, typically ranging from 40 to 50 GPa [21]. This low modulus of elasticity is crucial for preventing aseptic loosening of the implant due to differences in elastic modulus between the implant and the bone following Wolff’s law [22]. The Ti25Nb4Ta8Sn alloy is bioactive, non-cytotoxic, and serves as a suitable substitute for the widely used Ti6Al4V alloys in terms of mechanical properties and corrosion resistance [23].

This paper concentrates on the behavior of thin 3D-printed samples, aiming to experimentally assess their mechanical properties under various material compositions and modifications. The values obtained will help to calibrate numerical material model that will be able to simulate real-world experiments. Numerical tools can predict the response of gyroid structures based on different combinations of pore sizes and wall thicknesses [24].

## 2. Laboratory Experiments

In order to determine reliable material parameters of a numerical model describing materials applicable in biomedical implants, several samples were used in various test conditions. The following material parameters were determined: the tensile strength, Young’s modulus of elasticity, and porosity. Four groups of samples were used. Two alloys were considered, namely Ti25Nb4Ta8Sn and Ti6Al4V, and they were used without any post-processing or with surface etching and the removal of imperfectly fused powder grains through washing or with post-annealing. The fundamental groups are as follows:Group I—includes samples composed of Ti25Nb4Ta8Sn material without supplementary post-processing treatments.Group II—comprises Ti25Nb4Ta8Sn samples subjected to post-annealing to alleviate internal stresses.Group III—comprises specimens fabricated from Ti6Al4V material with annealing but without any additional post-processing treatments.Group IV—consists of Ti6Al4V samples subjected to annealing and etching of the surface and the removal of imperfectly fused powder grains through washing.

### 2.1. Preparation of Samples

The 3D printing samples of the groups I and II used Ti25Nb4Ta8Sn alloy powder (25 wt. % Nb, 4 wt. % Ta, 8 wt. % Sn, Ti balance) prepared by Advanced Metal Powders, Ltd. (Kouty, Czech Republic) The powder size and distribution is documented in Figure 1. It can be seen that the average powder size is around 50 μm with a distribution of 10–120 μm, which corresponds to the powder size distribution after atomization [25]. For the 3D printing of the beta titanium alloy, the selective laser melting method was used in collaboration with the Institute of Physics of the Academy of Sciences (Prague, Czech Republic). The samples were printed on a TruePrint 1000 with the following parameters: power 160 W, hatch distance 0.08 mm, energy 57.1 J/mm^3^. The heat treatment was set up in cooperation with UJP Prague, Ltd. (Prague, Czech Republic). The samples were annealed for 1 h at a temperature of 1000 °C followed by gradual cooling in the furnace. Preliminary tests demonstrated that Ti25Nb4Ta8Sn samples were almost completely dissolved under the applied HF–HNO_3_ etching conditions due to their high porosity; therefore, no etched group of this alloy was included in the experimental design.

The Groups III and IV of samples were produced using selective laser melting technique from Ti6Al4V titanium alloy (Rematitan CL—Ti6Al4V ELI Grade 23) with a maximum grain size of 63 μm. These samples were fabricated in collaboration with ProSpon, Ltd. (Kladno, Czech Republic) using an M2 Cusing machine. The 3D printing was conducted in an argon environment with 0.5% oxygen maintained within the welding chamber. Subsequently, the printed samples underwent thermal processing (gradual heating to 840 °C over 4 h, holding at 840 °C for 2 h and then followed by cooling to room temperature) in nearly vacuum conditions to relieve internal stresses.

### 2.2. Surface Etching and Porosity

For the purpose of mechanical testing and comparing the influence of post-processing treatments on the resulting mechanical properties, samples from group IV underwent surface etching. Surface etching aims to decrease the surface roughness and to remove partly unmelted surface powders. Surface treatment starts with 5 min degreasing in ethanol in ultrasonic bath. Samples were etched for 6 min in a solution of 20 mL HF, 200 mL HNO_3_ and 780 mL demineralized water in ultrasonic bath. Specimens were subsequently washed with demineralized water (in ultrasonic bath) and dried by hot air. Etching conditions were optimized based on our previous study [26]. Six minutes is the minimal time to achieve a surface with significantly removed partially melted particles [26].

Detailed microscopic analysis of the printed samples from groups III and IV did not confirm the presence of internal porosity, indicating that the samples can be considered as homogeneous materials. On the contrary, in the case of samples from groups I and II, significant internal porosity was identified, prompting further analysis. A cross-section of the Ti25Nb4Ta8Sn alloy sample in Figure 2 reveals incompletely fused powder particles and a high degree of porosity. The porosity of the samples was determined from discrete perpendicular sections of the test specimen, which were ground into P2500 roughness paper and then photographed on an SEM (Tescan Vega 3, Brno, Czech Republic). Image analysis was performed using FiJi software ImageJ, ver. 1.46r). The resulting porosity values range from 16.18 ± 0.88 % (for the thickest samples) to 19.2 ± 3.5 % (for the thinnest samples).

### 2.3. Mechanical Testing in Uniaxial Tension

Mechanical testing in uniaxial tension was conducted using a LiTeM machine (Italy) operating in controlled displacement mode. All dog bone specimens were vertically clamped within self-locking jaws and loaded along the longitudinal axis of the specimen. A loading rate of 0.04 mm/min was chosen in accordance with EN 10993 standards [27]. This standard, focusing on implant testing, was selected with the aim of utilizing the findings for the future design of intraosseous implant components.

In each group I–IV, three different thicknesses of dog bone-shaped samples were used, namely 500, 750, and 1000 μm with the total length of the samples being 50 mm (Figure 3 left). Six test specimens were evaluated for each thickness and group, resulting in a total of 72 samples. The minimum thickness of the samples was determined with respect to the technological capabilities of the 3D printer, while the maximum thickness was limited by the load capacity of the LiTeM testing machine (Ancona, Italy).

Thermal annealing is commonly applied in metal 3D printing to minimize internal stresses [14]. Therefore, all Ti6Al4V samples underwent this process. For the experimental material Ti25Nb4Ta8Sn, thermal treatment was conducted only on samples from group I to compare the mechanical properties of the base material without additional modifications. Samples of this material subjected to surface etching were not tested, as they were nearly completely dissolved during the etching process due to the high porosity of the material.

### 2.4. Micromechanical Testing

Micromechanical testing was performed on fragments of 3D-printed material made of Ti25Nb4Ta8Sn alloy. These fragments were cut using a slow-speed saw from dog bone samples that had not undergone uniaxial tensile testing. Fragments with sizes/lengths of approximately 2 × 2 × 0.75 mm were embedded in epoxy resin, and after curing, they were ground and polished for microhardness measurements and measured in three directions (Figure 3 right). The actual measurements were conducted using a CSM Instruments machine equipped with a Cube Corner Indenter. A controlled force mode was employed with a maximum load of 15 mN and a loading and unloading rate of 90 mN/min. The results were evaluated according to the standard Oliver & Phar methodology [28].

## 3. Results and Discussion

Laboratory tests served to obtain the metallographic and mechanical properties of the printed materials. The effect of annealing and surface etching on the material properties was studied.

### 3.1. Metallography and Surface Etching

In Figure 4A, compact areas without porosity, called molten pools, were visible in some parts of samples. The pools are typical for materials created by the selective laser melting. In addition to the pools, their boundaries can also be clearly seen. The molten pools are visible throughout the sample, but they are most noticeable in the areas where there are the fewest defects. Inside the molten pool boundaries, columns of cellular structure can be observed, whose orientation corresponds to the structure of the material.

Figure 4B shows a part of the sample where microstructural defects are already visible; in this case, it is probably a cold joint where perfect print bonding has not occurred. The higher weight loss observed for thinner samples is attributed to their larger surface-to-volume ratio and the higher proportion of partially melted powder particles at the surface, which are preferentially removed during etching. However, molten pool boundaries with cellular structure are still visible. In contrast, Figure 4C,D show more defective portions of the sample/microstructure. In the case of free surfaces, where unmelted powder particles are present, there is a more pronounced cellular/subcellular structure. At the same time, structural phenomena can be observed in the vicinity of unmelted powder grains that correspond to the microstructure of unmelted atomized powder (see Figure 4D).

Figure 5 illustrates the effect of etching on TiAlV alloys. It can be seen that after 6 min of etching, most of the partially melted powder spheres disappear from the surface. During the etching process, there was a relative weight decrease of 5%, 2%, and 1% for samples of 500 μm, 750 μm, and 1000 μm thickness, respectively.

The Beta alloy exhibited the highest roughness among all examined samples, with Sa = 28 μm and Sz = 241 μm. Its surface morphology was characterized by pronounced irregularities and deep valleys, indicating a highly uneven topography. Such a morphology is typical for additively manufactured materials, reflecting the layer-by-layer building process and incomplete melting in localized regions. The pronounced surface features may enhance mechanical interlocking or coating adhesion but could also serve as stress concentrators during mechanical loading.

In contrast, the Ti6Al4V alloy displayed a much smoother surface. The as-printed sample showed roughness values of Sa = 13 µm and Sz = 95 µm, while the etched surface exhibited reduced roughness, with Sa = 11 μm and Sz = 64 μm. The original surface morphology contained numerous partially melted and sintered powder particles, characteristic of the additive manufacturing process (Figure 6). These spherical particles contributed to the overall surface roughness and irregularity. After chemical etching, these partially fused particles were largely removed, resulting in a cleaner, more homogeneous surface with a finer microstructural relief. Consequently, the reduction in Sa and Sz values reflects both the removal of loose surface features and the exposure of the underlying microstructure. Overall, the comparison demonstrates that the surface treatment significantly affects the topography of the Ti–6Al–4V alloy. Etching effectively smoothened the surface by eliminating residual powder particles. These morphological differences, originating from the additive manufacturing process and subsequent surface modification, are expected to strongly influence the materials’ mechanical behavior.

### 3.2. Result of Mechanical Testing

The force–displacement curve from which the tensile strength and modulus of elasticity can be determined is one of the most important material characteristics. Uniaxial tests were performed in order to study the effect of the sample thickness on the force–displacement curves. Representative curves are depicted in Figure 7. Samples of Ti25Nb4Ta8Sn alloy were tested on six specimens without post-processing treatments (blue line) and on six specimens subjected to annealing to relieve internal stresses (red line). Ti6Al4V alloy samples were tested on six specimens with annealing only (blue line) and on six specimens after annealing and etching imperfectly welded dust spheres (green line).

The standard Ti6Al4V alloy exhibits significantly higher ultimate tensile strength values compared to the Ti25Nb4Ta8Sn alloy. This difference can be attributed to the significantly higher porosity in the beta-structured alloy. This high porosity also leads to a substantial reduction in strength after annealing, although there is not a notable change in Young’s modulus. Conversely, for the conventional alpha–beta alloy, there was no decrease in ultimate tensile strength for etched samples, but significant decreases in Young’s modulus were observed for the thinnest samples based on the graphs in Figure 7. As the thickness increases, the differences in the force–displacement curves diminish, suggesting that the quality of 3D printing plays a less critical role.

In Table 1, the recalculated values of ultimate tensile stress at fracture and the computed values of Young’s modulus allow for a comparison of the parameters of Ti25Nb4Ta8Sn with Ti6Al4V alloys obtained from samples with and without post-processing treatments.

The failure of samples from groups I and II in uniaxial tensile testing primarily occurred across grain boundaries, as evidenced by electron microscopy images (Figure 8). Additionally, a significant amount of imperfectly fused powder particles was observed, resulting in very low values of ultimate tensile strength and fracture forces. This phenomenon was only observed in the Ti25Nb4Ta8Sn alloy, which the authors attribute to improper printing settings. Optimizing the printing parameters will be the subject of further investigation and is not covered in this paper. However, for the purposes of numerical modeling and developing a particle model, this high porosity is advantageous as it allows for more detailed particle modeling. The morphology of the preceding beta grains and the resulting alpha phase variations significantly affect the anisotropic mechanical properties as seen in Ti6Al4V alloys [29].

The results of the micromechanical evaluation confirm the assumptions from previous experiments [30] that the material behaves non-isotropically and that its mechanical strength will depend on the printing direction. The layered structure (Figure 8, left) corresponds more to a material with isotropic behavior, as confirmed by nanoindentation. For all indentation directions, three matrices of indentations sized 3 × 3 were performed at the left edge, center, and right edge of the tested sample. No significant differences in micromechanical properties between the edge and center of the sample were observed. The values of hardness, $H_{IT}$, and modulus of elasticity, $E_{IT}$, for the defined indentation directions depicted in Figure 3 are presented in Table 2, and the differences in the variables in various directions are substantial. It is important to note that these values are influenced by significant sample porosity and possible elastic interlayering of individual layers within the resin matrix.

In comparison with the literature [23], where the cast structure of beta titanium alloy Ti25Nb4Ta8Sn had 0.2% offset yield stress $\sigma_{0.2}$ of 435 MPa and a reduced elastic modulus $E_{r}$ of 61 GPa [20], in this case, the yield stress $\sigma_{0.2}$ in the axial direction reached 780 MPa.

## 4. Conclusions

Two generations of titanium alloys for biomedical use were investigated—the conventional Ti6Al4V and the novel beta titanium alloy Ti25Nb4Ta8Sn. Samples were produced by selective laser melting (SLM) and were subsequently treated by annealing and surface etching. Various specimen thicknesses were tested under uniaxial tension to obtain force–displacement diagrams. The Ti6Al4V alloy exhibited higher tensile strength compared to Ti25Nb4Ta8Sn. Microstructural analysis revealed features such as porosity and molten pool boundaries, while surface etching efficiently removed partially melted powder particles. The micromechanical evaluation indicated anisotropic behavior, with mechanical properties depending on the printing direction.

Certain limitations of this study should be noted. Etching of beta titanium alloys remains challenging, as their intrinsic porosity often leads to excessive dissolution during standard chemical treatment. This issue requires an optimization of surface modification procedures specifically tailored for beta-phase materials. The experimental data presented here will serve as a foundation for subsequent numerical modeling and for extending the research toward fatigue testing, which is essential for evaluating the long-term performance of beta titanium alloys in biomedical applications.

The Ti25Nb4Ta8Sn alloy combines proven biocompatibility with excellent mechanical compatibility. Its low Young’s modulus (∼60 GPa), close to that of natural bones, helps reduce stress shielding and supports implant stability. Moreover, the alloy is non-toxic and free from potentially harmful elements such as aluminum and vanadium, making it a promising candidate for intraosseous implants. Further verification of the Ti25Nb4Ta8Sn alloy as a suitable biomaterial must be carried out through in vitro and in vivo experiments, which are currently in progress in project No. NW24-1000118 (The Czech Health Research Council).

## Figures and Tables

**Figure 1 materials-18-05008-f001:**
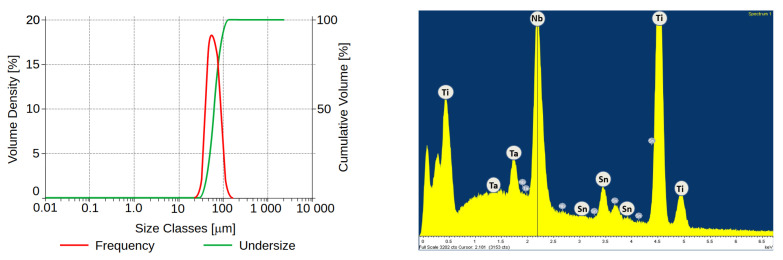
Distribution of Ti25Nb4Ta8Sn alloy powder used for selective laser melting printing; red graph shows frequency and green shows undersize (**left**). Illustration of Ti25Nb4Ta8Sn alloy powder size and distribution, Energy–Dispersive Spectroscopy analysis of powder chemical composition (**right**).

**Figure 2 materials-18-05008-f002:**
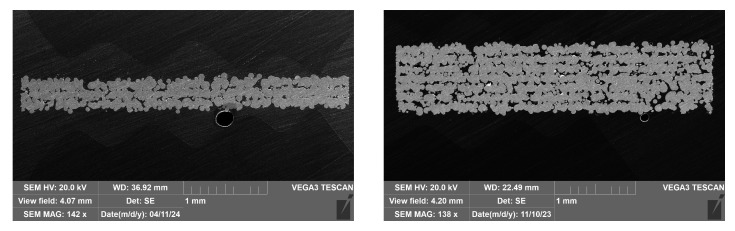
Example of the Ti25Nb4Ta8Sn SEM image for porosity determination. Thickness 500 μm (**left**) and 1000 μm (**right**).

**Figure 3 materials-18-05008-f003:**
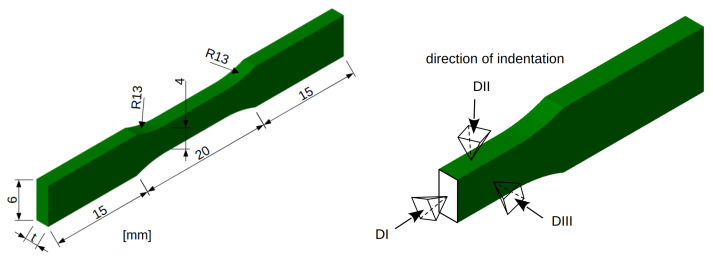
Geometry of dog bone-shaped test specimens for uniaxial tensile testing, with a thickness t (**left**). Schematic representation of the orientations of the indentation directions in the CSM Instruments apparatus (**right**).

**Figure 4 materials-18-05008-f004:**
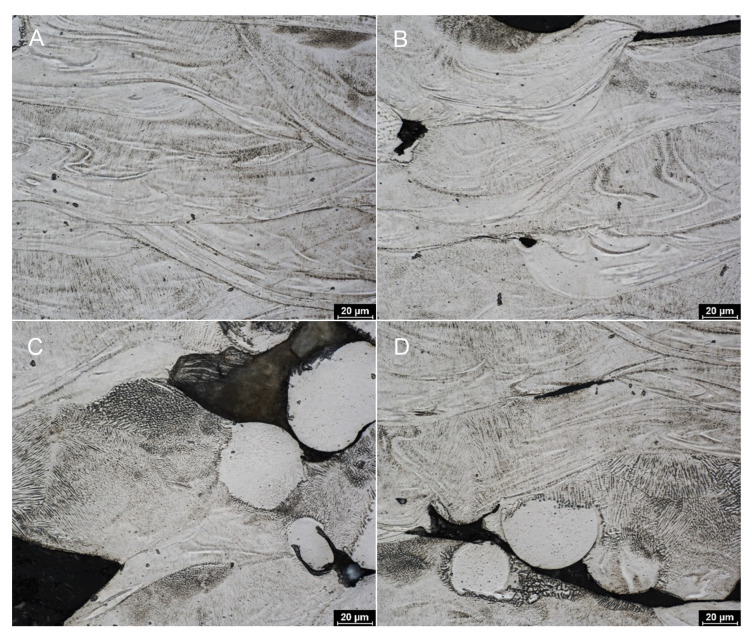
Microstructure of printed Ti25Nb4Ta8Sn alloy—Group I. (**A**) Molten pools with visible boundaries and columnar cellular structures aligned with the heat flow direction. (**B**) Cold joint region showing incomplete layer fusion and local material loss. (**C**,**D**) Defective areas with increased porosity, unmelted powder particles, and structural features resembling the original atomized powder, indicating insufficient local laser energy transfer.

**Figure 5 materials-18-05008-f005:**
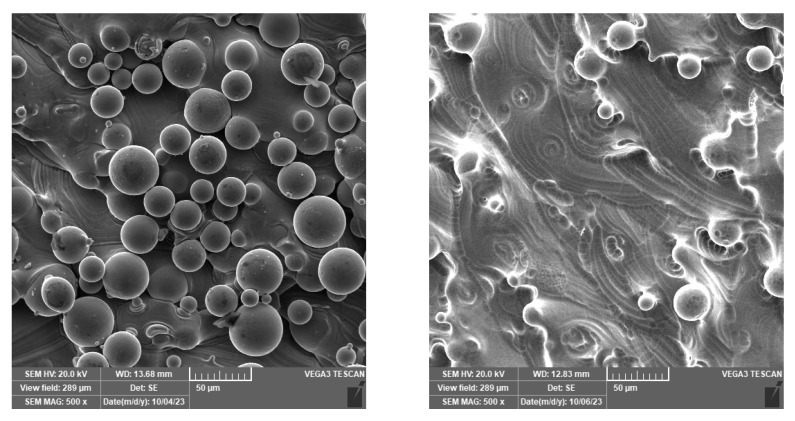
The surface of as-printed (**left**) and etched (**right**) TiAlV samples.

**Figure 6 materials-18-05008-f006:**
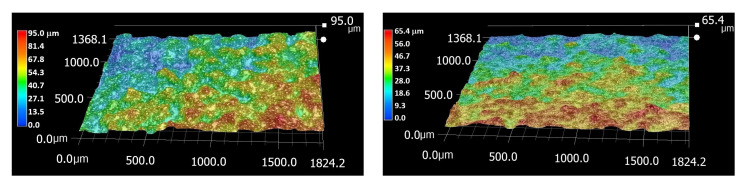
Surface roughness of Ti6Al4V material captured by a 3D microscope. On the **left**: the original surface; on the **right**: the surface after chemical etching.

**Figure 7 materials-18-05008-f007:**
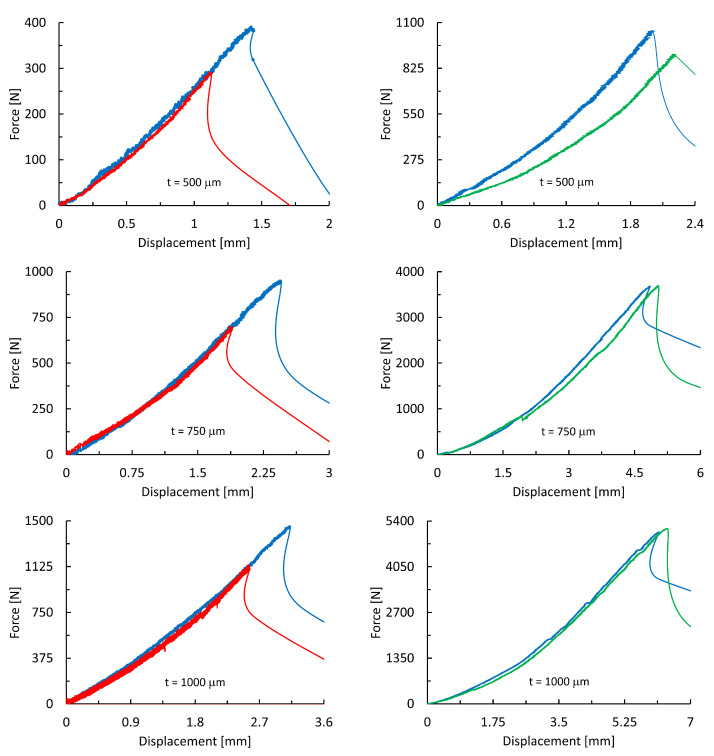
The force–displacement relationship from uniaxial tensile testing for materials Ti25Nb4Ta8Sn (**left column**) and Ti6Al4V (**right column**).

**Figure 8 materials-18-05008-f008:**
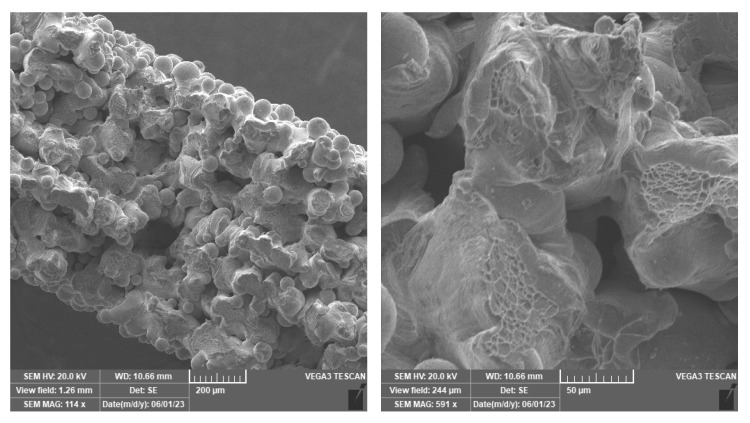
An electron microscope image of the fracture surfaces of a sample with the thickness of *t* = 750 μm.

**Table 1 materials-18-05008-t001:** Average values of maximum force, ultimate tensile strength, and Young’s modulus of elasticity for different types of samples.

		Ti25Nb4Ta8Sn		Ti6Al4V	
		**Without**	**With**	**Without**	**With**
**t [μm]**		**Annealing**	**Annealing**	**Etching**	**Etching**
	$F_{max}$ [N]	408 ± 12	208 ± 30	1009 ± 43	760 ± 85
500	$R_{m}$ [MPa]	204 ± 6	135 ± 15	505 ± 22	380 ± 75
	*E* [MPa]	3620 ± 108	3730 ± 410	7910 ± 320	6780 ± 630
	$F_{max}$ [N]	889 ± 38	674 ± 86	3570 ± 129	3571 ± 106
750	$R_{m}$ [MPa]	296 ± 13	225 ± 29	1190 ± 83	1190 ± 62
	*E* [MPa]	3810 ± 152	3970 ± 460	7930 ± 310	7790 ± 234
	$F_{max}$ [N]	1443 ± 69	1011 ± 101	4579 ± 180	4959 ± 133
1000	$R_{m}$ [MPa]	361 ± 17	253 ± 25	1145 ± 145	1247 ± 46
	*E* [MPa]	3940 ± 190	4184 ± 420	7950 ± 390	8040 ± 280
		Group I	Group II	Group III	Group IV

**Table 2 materials-18-05008-t002:** Hardness, $H_{IT}$, and modulus of elasticity, $E_{IT}$, computed as the arithmetic mean from the indents in a specific direction. The hardness and modulus of elasticity were evaluated according to the Oliver & Pharr method [28].

	Direction of Nanoindentation		
	**DI**	**DII**	**DIII**
$H_{IT}$ [MPa]	3212 ± 189	3376 ± 219	2336 ± 188
$E_{IT}$ [GPa]	52.2 ± 4.8	75.9 ± 3.3	16.1 ± 0.8

## Data Availability

The data presented in this study are available on request from the corresponding author. Their disclosure is restricted due to ongoing intellectual property protection proceedings.

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
