# Peer review of "Effect of Sample Thickness and Post-Processing on Mechanical Properties of 3D-Printed Titanium Alloy"

_materials, 2025, doi:10.3390/ma18215008_

Round 1

Reviewer 1 Report

Comments and Suggestions for Authors

This article focuses on biomedical 3D-printed titanium alloys, with Ti25Nb4Ta8Sn and Ti6Al4V as the research objects. Samples were prepared via selective laser melting (SLM), divided into four groups for treatment. Porosity, uniaxial tensile properties, and micromechanical properties were tested, and the research results were elaborated. The overall logical expression of the article is accurate, but there are still some questions, as detailed below:

Question 1: The abstract fails to highlight core findings, only mentioning “measured parameters” and lacking conclusive information. Revision and supplementation are recommended.

Question 2: Line 14 contains the character “)”, while the preceding text does not have the corresponding “(”. This does not comply with paper standards, and deletion is recommended.

Question 3: The introduction does not clearly identify gaps in existing research, resulting in insufficient emphasis on the necessity of this study. Supplementary discussion on the necessity of this research is recommended.

Question 4: The reason why no etching group was set up for Ti25Nb4Ta8Sn was not explained at the experimental design stage. It is recommended to clarify this and supplement the basis for experimental design in Section 2.1 “Sample Preparation”.

Question 5: HIT and EIT in Table 2 should be clearly defined in the table title.

Question 6: Lines 168–171 only describe that “the weight loss of Ti6Al4V after etching is 5% (500μm), 2% (750μm), and 1% (1000μm)”, without explaining “why the thinner the sample, the greater the weight loss”.

Question 7: Lines 183–185 state that “high porosity leads to a significant decrease in strength after annealing”, but the data in Table 1 shows an increase. It is necessary to confirm whether this is a data arrangement error.

Question 8: The conclusion does not specify the limitations and shortcomings of the study, which should be supplemented.

Question 9: In Line 87, there is no space between the number and the unit in “57.1J/mm³” (should be “57.1 J/mm³”).

Question 10: In Line 89, there is no space between the number and the unit in “1000℃”(should be “1000 °C”).

Question 11: Entries 26–32 in the references are incomplete, and revision is recommended.

After the above revisions, the data credibility, logical rigor, format standardization, and field adaptability of the target paper will be significantly improved, better meeting the publication requirements of journals in the field of biomedical 3D printing.

Author Response

Review 1

Question 1: The abstract fails to highlight core findings, only mentioning “measured parameters” and lacking conclusive information. Revision and supplementation are recommended.

Response 1: Thank you for this valuable comment. The abstract has been revised to clearly emphasize the key findings and numerical results. The revised version now summarizes the main experimental outcomes, including the comparison of tensile strength and elastic modulus for Ti25Nb4Ta8Sn and Ti6Al4V alloys, as well as the observed effects of annealing and etching.

Question 2: Line 14 contains the character “)”, while the preceding text does not have the corresponding “(”. This does not comply with paper standards, and deletion is recommended.

Response 2: Thank you for your comment – it has been corrected.

Question 3: The introduction does not clearly identify gaps in existing research, resulting in insufficient emphasis on the necessity of this study. Supplementary discussion on the necessity of this research is recommended.

Response 3: Thank you for your comment. Additional information regarding the need for the proposed solution has been added to the Introduction (lines 32–38 and 58–62). Furthermore, references 4–8 have been included to support this section.

Question 4: The reason why no etching group was set up for Ti25Nb4Ta8Sn was not explained at the experimental design stage. It is recommended to clarify this and supplement the basis for experimental design in Section 2.1 “Sample Preparation”.

Response 4: We thank the reviewer for this valuable comment. The reason why no etching group was established for the Ti25Nb4Ta8Sn alloy is that preliminary tests revealed a very high degree of internal porosity in this material. When exposed to the same HF–HNO₃ etching solution used for Ti6Al4V, the Ti25Nb4Ta8Sn samples were almost completely dissolved within a few minutes. This was due to the open porous structure and the chemical reactivity of the β-titanium alloy in this environment. Consequently, the etching process could not be optimized without compromising the integrity of the samples. For this reason, no etched group was included in the experimental design.
To clarify this point, the following sentence has been added at the end of Section 2.1 “Sample Preparation”:

“Preliminary tests demonstrated that Ti25Nb4Ta8Sn samples were almost completely dissolved under the applied HF–HNO₃ etching conditions due to their high porosity; therefore, no etched group of this alloy was included in the experimental design.”

Question 5: HIT and EIT in Table 2 should be clearly defined in the table title.

Response 5: Thank you for the comment. The definitions of HIT and EIT have been added to the table caption, and a reference to the Oliver and Pharr method has been included.

Question 6: Lines 168–171 only describe that “the weight loss of Ti6Al4V after etching is 5% (500μm), 2% (750μm), and 1% (1000μm)”, without explaining “why the thinner the sample, the greater the weight loss”.

Response 6: We appreciate the reviewer’s observation. The greater weight loss in thinner Ti6Al4V samples after etching is primarily related to the higher specific surface area to volume ratio. In thinner samples, the proportion of surface material—where partially melted powder particles are most abundant—is larger compared to the bulk. Since etching removes these surface particles and the surrounding oxide layer, the relative mass loss increases as the sample becomes thinner.
The following clarifying explanation has been added to Section 3.1 (Lines 181-184):

“The higher weight loss observed for thinner samples is attributed to their larger surface-to-volume ratio and the higher proportion of partially melted powder particles at the surface, which are preferentially removed during etching.”

Question 7: Lines 183–185 state that “high porosity leads to a significant decrease in strength after annealing”, but the data in Table 1 shows an increase. It is necessary to confirm whether this is a data arrangement error.

Response 7: Thank you for pointing this out. This was an error in Table 1 caused by a mix-up of columns. The table has been corrected accordingly.

Question 8: The conclusion does not specify the limitations and shortcomings of the study, which should be supplemented.

Response 8: We thank the reviewer for this constructive recommendation. The Conclusion section has been revised and expanded to include the main limitations of this study. Additional information regarding the safety and biocompatibility of β-titanium alloys in medical applications has been incorporated, along with an outline of future research directions for this advanced alloy.

Question 9: In Line 87, there is no space between the number and the unit in “57.1J/mm³” (should be “57.1 J/mm³”).

Response 9: A space between the value and the unit was already included, but to ensure proper formatting I have replaced it with a non-breaking space so that the value and the unit cannot be separated at the end of a line.

Question 10: In Line 89, there is no space between the number and the unit in “1000℃”(should be “1000 °C”).

Response 10: A space between the value and the unit was already included, but to ensure proper formatting I have replaced it with a non-breaking space so that the value and the unit cannot be separated at the end of a line.

Question 11: Entries 26–32 in the references are incomplete, and revision is recommended.

Response 11: Thank you for pointing this out. The cited references remained in the text by mistake (they were part of the template). This has now been corrected in the revised version.

Reviewer 2 Report

Comments and Suggestions for Authors

Many thanks for submitting your article to Materials for publication, it is welcome. After a deep analysis on your paper, I would like to comment that although the area of investigation could be of interest for novel metallurgists, some topics on your paper could be improved before acceptation. You can find it in boxes inside the main body of your text, where the main questions appear.

Author Response

Review 2

Could you please identify the main features of these micrographs? i.e. porosity voids, lines of flow, etc.

Response 1: Thank you for your comment. A shortened description of the microstructure has been added to the figure caption. A more detailed explanation is provided in the first two paragraphs of Section 3.1.

As porosity is the main drawback of laser printed pieces of the T125Nb4Ta8Sn alloys, could you add a table where measurements of the percentage of porosity can be seen?

Response 2: Thank you for your suggestion. The porosity values are already provided on pages 138 and 139. We believe that adding an additional table would not significantly improve the clarity of the manuscript. However, we fully agree that porosity is one of the key parameters. Therefore, in our upcoming work, we plan to compare the effect of varying 3D printing parameters on the resulting porosity of the Ti25Nb4Ta8Sn  alloy.

Optimizing the printing parameters will be the subject of further investigation and is not part of this paper. However, for the purposes of numerical modeling and developing a particle model, this high porosity is advantageous as it allows for more detailed particle modeling. In comparison with the literature [17], where the cast structure of beta titanium alloy Ti25Nb4Ta8Sn has 0.2% offset yield stress, σ0.2, of 435 MPa and reduced elastic modulus, Er, of 61 GPa [14].

This explanation lacks of argument

Response 3: Thank you for pointing this out. This was an error, and the sentence “In comparison with the literature [17], where the cast structure of the beta titanium alloy Ti25Nb4Ta8Sn has a 0.2% offset yield stress, σ₀.₂, of 435 MPa and a reduced elastic modulus, Eᵣ, of 61 GPa [14]” has been removed from this section of the text.

It is better that this graph can be presented before the conclusions paragraph

Figure 7 and also Table 2 should be better presented before the conclusions paragraph.

Response 4: Figures 6 and 7, as well as Table 2, are placed in the LaTeX document before the Conclusion section. However, the template parameters automatically position figures according to optimal typographic layout. By adding additional text, the figures have now been shifted to appear at least before the Conclusion chapter.

What is the meaning of this kind of reference´s presentation?

Response 5: Thank you for pointing this out. The cited references remained in the text by mistake (they were part of the template). This has now been corrected in the revised version.

Reviewer 3 Report

Comments and Suggestions for Authors
  1. Lines 18-19 - please write how precise can be 3D printed structure. Please refer to specific values.
  2. Line 23 - what do authors mean by "complex" structure? When structure is complex and when not?
  3. Line 41 - Are the authors considering Ti6Al4V for medical applications or Ti6Al4V ELI? This makes a significant difference in terms of chemical composition and subsequent biomedical applications. At this point, more references to the literature comparing these two alloys would be useful.
  4. The Authors also inform that Ti6Al4V can provide pottential toxicity. Please refer to more literature in that point.
  5. The introduction is interesting and well-written, but requires further expansion and expansion. While the authors describe the characteristics of 3D printing and titanium alloys, they omit the measurement aspects of the fabricated structures. The authors, of course, present the results of their own research, but they write little about it in the literature review.
  6. Fig. 4 - it is worth describe components of the microstructure. Add descriptions in the figures.
  7. Fig. 2  - Changing the 3D printing parameters would likely change the porosity. So, did the authors anticipate varying the printing parameters to achieve the desired porosity?
  8. Are surface aspects also important to the authors? It seems that surface parameters, such as roughness, are crucial for implants. Did the authors provide these results?
  9. Surface texture is important. Several classic methods for measuring texture can be used, including confocal optical microscopy, interferometric microscopy, and focus variation, as well as innovative methods for measuring surface texture, such as using an elastomeric touch sensor that imprints the surface texture and enables its mapping - please refer to: https://doi.org/10.3390/ma18132939 The authors should also include results of surface topographic characterization in their studies. Or at least add a paragraph describing the significance of surface texture and the methods for its measurement.
  10. In the conclusions the Authors should add information about Ti25Nb4Ta8Sn safety in biomedical applications, especially when they want apply this alloy in human body. 

Author Response

Review 3

Lines 18-19 - please write how precise can be 3D printed structure. Please refer to specific values.

Response 1: Thank you for your comment. Defining clear limits of accuracy in metal 3D printing is very challenging, as it always depends on the quality of the input materials and machine calibration. In general, with the equipment used, the achievable accuracy is within ±0.05 mm. A description of this statement has been added to the Introduction (lines 24-29).

Line 23 - what do authors mean by "complex" structure? When structure is complex and when not?

Response 2: We consider a complex geometric structure to be one that is defined implicitly and whose parameters can be modified through adjustments to its defining equation (in our case, the gyroid equation (1)). Changing parameters such as the wall thickness-to-pore size ratio is essential for different types of implants (dental, hip, patient-specific, etc.) and allows the density to be optimized according to the bone density at the implantation site.
In contrast, simple structures such as Rhombic or Dodecahedral types have a predefined geometry, where the ratio of wall thickness to pore size or the curvature can be modified only through graphical editing rather than mathematical formulation.

Line 41 - Are the authors considering Ti6Al4V for medical applications or Ti6Al4V ELI? This makes a significant difference in terms of chemical composition and subsequent biomedical applications. At this point, more references to the literature comparing these two alloys would be useful.

Response 3: Thank you for pointing this out. Additional information has been included in the text (lines 32-38 and 113). Throughout the manuscript, we focus on the Ti6Al4V ELI alloy, which is used by the implant manufacturer ProSpon, Ltd. The manufacturer produces the test specimens for us on demand, following the same standards as those applied in the actual implant production. A certified Ti6Al4V powder, Rematitan CL (ELI Grade 23), is used for this purpose.

The Authors also inform that Ti6Al4V can provide pottential toxicity. Please refer to more literature in that point.

Response 4: Thank you for your comment. Additional information has been added to the Introduction (lines 32–38 and 58-62). Furthermore, references 4–8 have been included to support this section.

The introduction is interesting and well-written, but requires further expansion and expansion. While the authors describe the characteristics of 3D printing and titanium alloys, they omit the measurement aspects of the fabricated structures. The authors, of course, present the results of their own research, but they write little about it in the literature review.

Response 5: Thank you for your comment. Additional information regarding 3D printing accuracy, toxicity, and the general motivation for addressing this issue has been added to the Introduction. Based on our experience, measuring the final printed samples is very challenging, and the obtained values often do not fully correspond to the geometrical model (small deviations, such as variations in sample wall thickness, frequently occur). This topic is partially discussed in the previous work by Jíra, A., et al. (2021), Mechanical Properties of Porous Structures…, Materials, 14(16), 4592, cited as reference [13]. In practice, the deviations from theoretical models are particularly significant in thin samples, which is why we focus on these in our study.

Fig. 4 - it is worth describe components of the microstructure. Add descriptions in the figures.

Response 6: Thank you for your comment. A shortened description of the microstructure has been added to the figure caption. A more detailed explanation is provided in the first two paragraphs of Section 3.1.

Fig. 2  - Changing the 3D printing parameters would likely change the porosity. So, did the authors anticipate varying the printing parameters to achieve the desired porosity?

Response 7: The 3D printing parameters play a crucial role in determining the final porosity of the structure. In the case of the Ti25Nb4Ta8Sn  alloy, we plan to optimize the printing parameters in our follow-up work — several test samples with different printing settings will be fabricated and subsequently evaluated in terms of porosity and microstructure. For the Ti6Al4V alloy, however, we do not anticipate any modification of the printing methodology, as the printing is carried out by an industrial partner following certified procedures for medical applications.

Are surface aspects also important to the authors? It seems that surface parameters, such as roughness, are crucial for implants. Did the authors provide these results?

Surface texture is important. Several classic methods for measuring texture can be used, including confocal optical microscopy, interferometric microscopy, and focus variation, as well as innovative methods for measuring surface texture, such as using an elastomeric touch sensor that imprints the surface texture and enables its mapping - please refer to: https://doi.org/10.3390/ma18132939 The authors should also include results of surface topographic characterization in their studies. Or at least add a paragraph describing the significance of surface texture and the methods for its measurement.

Response 8: Thank you for this valuable comment. We agree that surface characteristics are crucial for implant applications. Therefore, we have added a new section describing the surface roughness and morphology of both alloys, including quantitative roughness values (Sa, Sz) and their correlation with the additive manufacturing process and post-processing treatment (lines 194-214 and Fig. 6). This addition highlights the significance of surface texture and its potential impact on mechanical performance.

In the conclusions the Authors should add information about Ti25Nb4Ta8Sn safety in biomedical applications, especially when they want apply this alloy in human body.

Response 9: We thank the reviewer for this constructive recommendation. The Conclusion section has been revised and expanded to include the main limitations of this study. Additional information regarding the safety and biocompatibility of β-titanium alloys in medical applications has been incorporated, along with an outline of future research directions for this advanced alloy.

Round 2

Reviewer 1 Report

Comments and Suggestions for Authors

The authors have revised the manuscript according to the comments. Therefore, this manuscript is acceptable in its current form.

Author Response

We sincerely thank the reviewer for the positive evaluation and for confirming that the manuscript is acceptable in its current form. 

Reviewer 2 Report

Comments and Suggestions for Authors

Dear authors, I have checked the revised form of your paper, but despite you have addressed most of the changes requested, still there are three items that you must add, as follows:

Figure 9 is still small enough for you to enlarge it.

Figure 16 is the same request.

You do not include most of the references, please check.

Author Response

Thank you for your review and evaluation of our manuscript. We would like to kindly ask for clarification regarding some of the comments - the article ends with Figure 8. Therefore, we are not certain what is meant by the comment about the insufficient quality of Figures 9 and 16, as no such figures appear in the manuscript.

Nevertheless, we have revised Figure 6 to improve its readability. In addition, we have carefully checked all cited references, and each source mentioned in the reference list is properly cited in the text.